# Can We Predict Differentiated Thyroid Cancer Behavior? Role of Genetic and Molecular Markers

**DOI:** 10.3390/medicina57101131

**Published:** 2021-10-19

**Authors:** Rita Niciporuka, Jurijs Nazarovs, Arturs Ozolins, Zenons Narbuts, Edvins Miklasevics, Janis Gardovskis

**Affiliations:** 1Department of Surgery, Riga Stradins University, Pilsonu Street 13, LV-1002 Riga, Latvia; arturs.ozolins@rsu.lv (A.O.); zenons.narbuts@rsu.lv (Z.N.); janis.gardovskis@rsu.lv (J.G.); 2Department of Surgery, Pauls Stradins Clinical University Hospital, Pilsonu Street 13, LV-1002 Riga, Latvia; 3Department of Pathology, Pauls Stradins Clinical University Hospital, Pilsonu Street 13, LV-1002 Riga, Latvia; jurijs.nazarovs@stradini.lv; 4Institute of Oncology, Riga Stradins University, Pilsonu Street 13, LV-1002 Riga, Latvia; edvins.miklasevics@rsu.lv

**Keywords:** thyroid cancer, molecular markers, genetic testing, thyroid cancer clinicopathological features

## Abstract

Thyroid cancer is ranked in ninth place among all the newly diagnosed cancer cases in 2020. Differentiated thyroid cancer behavior can vary from indolent to extremely aggressive. Currently, predictions of cancer prognosis are mainly based on clinicopathological features, which are direct consequences of cell and tissue microenvironment alterations. These alterations include genetic changes, cell cycle disorders, estrogen receptor expression abnormalities, enhanced epithelial-mesenchymal transition, extracellular matrix degradation, increased hypoxia, and consecutive neovascularization. All these processes are represented by specific genetic and molecular markers, which can further predict thyroid cancer development, progression, and prognosis. In conclusion, evaluation of cancer genetic and molecular patterns, in addition to clinicopathological features, can contribute to the identification of patients with a potentially worse prognosis. It is essential since it plays a crucial role in decision-making regarding initial surgery, postoperative treatment, and follow-up. To date, there is a large diversity in methodologies used in different studies, frequently leading to contradictory results. To evaluate the true significance of predictive markers, more comparable studies should be conducted.

## 1. Introduction

Thyroid cancer (TC) is ranked in ninth place among all the newly diagnosed cancer cases in 2020, with an estimated incidence of 6.6 cases per 100,000 and mortality of 0.43 cases per 100,000 [1].

Differentiated thyroid cancer (DTC), which includes papillary cancer (PTC) and follicular cancer (FTC), comprises >90% of all thyroid cancer cases [2]. 

The etiology of TC is not completely clear, but it is known that diverse genetic and epigenetic changes are the driving forces of thyroid cancer development. Genetic and molecular alterations are noticed in the vast majority (>96%) of DTCs. The thyroid cancer progression is driven by the accumulation of genetic and epigenetic alterations with the corresponding progressive uncontrolled activity of various signaling pathways (e.g., MAPK and PI3K–AKT). Further, alterations of cell homeostasis, proliferation, differentiation, migration, and apoptosis control are induced in response to abnormal signaling. These processes are mediated directly or indirectly through numerous secondary molecule derangements. Activation and further cooperation of genes and molecules, along with the interaction between the tumor and its surrounding environment, are core aspects of the development and progression of DTC [3,4,5]. 

The vast majority of DTC have relatively inert biological behavior and an overall favorable prognosis. More than 90% of PTC patients have a life expectancy of over 10 years. However, DTC can occasionally obtain more aggressive features, as 2% of patients already have distant metastases at the time of diagnosis [6], and up to 20% will suffer from regional or local tumor recurrence [7]. Low-risk patients have a recurrence rate of <1%, but it could exceed >50% in high-risk. Risk categories are determined based on clinical and pathologic features, such as the specific histologic variant of TC, size of the primary tumor, degree of invasion, e.g., extrathyroidal extension (ETE), vascular invasion, presence, number, and size of locoregional lymph node metastases (LNM), etc. [8]. DTC’s outcomes can significantly differ, from complete remission in the vast majority of cases to rapid progression and death, suggesting large heterogeneity among cancers. 

Although several scoring systems are suggested to identify low-risk and high-risk thyroid carcinomas, they are mainly based on clinicopathological data. There is currently no supreme criterion that would facilitate defining optimal treatment for each cancer case individually. On the one hand, recognizing high-risk patients at the time of diagnosis would be essential to develop new management strategies and improve follow-up; on the other hand, it would help avoid overtreatment in low-risk patients. As representatives of cell processes and oncogenesis, genetic and molecular markers could be highly useful for additional risk assessment and identification of new treatment targets.

This review focuses on describing results of studies regarding the development of DTC aggressive features, recurrence, and overall survival. To achieve this, genetic aberrations, disarrangements of cell cycle control molecules, estrogen receptors expression patterns, intensity of epithelial-mesenchymal transition molecules, matrix metalloproteinases, hypoxia, and neovascularization molecules expressions were analyzed.

## 2. Genetic Aberrations Markers

Testing genetic mutations associated with TC may enhance the diagnostic value of pre-operative cytologic examination and post-operatively the predictability of adverse clinical outcomes.

Thyroid cancer is a genetically simple disease with a relatively low number of mutations in each tumor. Some of the alterations are so-called “driver” mutations that initiate normal cell transition into cancerous. Others are called “passenger” mutations—consequences of carcinogenesis. Alterations in BRAF, *RAS*, *RET*/PTC, and *TERT* promoters are assumed as “driver” changes and could be used as diagnostic markers for cancer development [9].

### 2.1. BRAF^V600E^

The *BRAF* gene is altered in 66.54% of PTC patients; *BRAF^V600E^*, in particular, is presented in 63.16% of these cases [10].

Significant heterogeneity has been noticed among literature data regarding the association of *BRAF^V600E^* mutation and aggressive PTC features. In several large meta-analyses and cohort studies, mutation strongly correlated with gender, LNM, absence of capsule, ETE, tumor multifocality, advanced stage, and specific subtypes of PTC. However, there were contradicting data regarding tumor size, patient age, and the presence of vascular invasion [11,12,13,14,15]. *BRAF^V600E^* is additionally suggested as a predictive marker of PTC recurrence or persistence, decreased efficacy of radioactive iodine (RAI) treatment [16], and reduced overall survival [12,13,17]. At the same time, Ahn et al. and Yan et al. presented no correlation between *BRAF^V600E^* and any of the PTC features [18,19]. Li et al. suggested that the geographical location and tumor stage should be considered when the impact of a BRAF mutation on the cancer outcome is assessed [20].

In papillary microcarcinomas (PTMC), *BRAF^V600E^* mutation is presented in 48–51% of cases. It is associated with larger tumor size, multifocality, ETE, LNM [21,22], and recurrence of PTMC [23].

*BRAF^V600E^* is suggested to have diagnostic value in the determination of cancer in fine-needle aspiration biopsy (FNAB) material. It improves diagnostic accuracy and reduces false-negative results. *BRAF^V600E^* positive were 23% of the indeterminate nodes [24]. Adeniran et al. reported that *BRAF^V600E^* sensitivity, specificity, positive predictive value, and negative predictive values for PTC in indeterminate nodules were 59.3%, 100%, 100%, and 65.6%, respectively [25]. On average, diagnostic sensitivity increased from 81.4 to 87.4%, but false-negative results decreased from 8% to 5.2% [24]. The opposite opinion was presented in the meta-analysis of Trimboli et al., where the *BRAF^V600E^* mutation was found only in 5% of the indeterminate nodules; therefore, diagnostic probability and cost-effectiveness are relatively low [26].

### 2.2. TERT Mutation

Telome*RAS*e reverse transcriptase (*TERT*) is the protein subunit of telome*RAS*e, which adds telomeres at the end of chromosomes. At each mitosis, telomeres become shorter, denoting cell ageing. When telomeres become critically short, cell apoptosis is induced. *TERT* promoter mutation is among the most recognized markers associated with aggressive TC phenotypes [27,28].

The average frequency of *TERT* promoter mutations in DTC, PTC, FTC, and dedifferentiated tumors is 10.9%, 10.6%, 15.1%, and 40%, respectively. In FTC and PTC, *TERT* promoter mutation is significantly associated with patients’ age, cancer size, presence of ETE, vascular invasion, LNM, distant metastases, advanced tumor stage, persistence, or recurrence, and disease-specific mortality [29,30,31]. Despite several similarities with previous studies, Liu and Xing’s meta-analysis reported a lack of correlation with lymph node metastasis. However, there was a strong trend of higher prevalence of LNM in mutation-positive patients [32]. In addition, Liu et al. found no correlation between *TERT* mutation and ETE, multifocality, and vascular invasion [33].

### 2.3. Combination of TERT and BRAF^V600E^ Mutation 

*TERT* promoter mutation frequently is found in combination with *BRAF^V600E^* mutation [32,34]. A combination of *BRAF^V600E^* and *TERT* has been noticed in all thyroid cancer histotypes, but it has specific significance in PTC. A significant correlation was found with patients age, ETE, LNM, distant metastasis, disease stage, tumor recurrence, mortality, and thyroid capsule invasion [35,36].

A combination of *TERT* and *BRAF^V600E^* mutations could be beneficial in the determination of cancers in intermediate nodules in FNAB. *TERT* promoter mutations alone have 100% specificity and only 7.0% sensitivity due to its low overall presence in thyroid cancers. When *TERT* mutations were assessed in conjunction with BRAF ^V600E^, the diagnostic specificity remained 100%, but sensitivity increased to 38.0%. Similarly, BRAF mutation sensitivity increased to 32.6%, according to Liu et al. [37]. 

### 2.4. RAS Mutation

*RAS* mutation represents the second most commonly identified genetic mutation in thyroid cancer. 

*RAS* is detected in 48% of follicular adenoma (FA), 57% of FTC, and only in 21% of PTC cases. *RAS*-positive DTC patients have a higher risk for *RAS*-mediated tumor dedifferentiation, development of distant metastases, recurrence, and worse life expectancy. However, association with clinicopathological manifestation is controversial [38,39]. *RAS* mutation-positive patients have a 2.9 times higher risk for disease-specific death than without *RAS* mutation [40].

*RAS* mutation status is suggested to have significant diagnostic utility for indeterminate nodules in FNAB. The predictive value for cancer ranges between 74% and 88%. *RAS* positivity also frequently correlates with the development of bilateral cancers. Thus, in *RAS* positive indeterminate nodules, more aggressive initial therapy should be considered—total thyroidectomy to reduce the necessity for completion reoperations that increase risks and expenses [39,41].

On the contrary, Povoa et al. and Xing et al. found no relation between mutation and poorer prognosis. Most patients with *RAS* mutation alone had a stage I diagnosis, usually presenting excellent responses to therapy, and were free of disease at the end of follow-up [42,43].

### 2.5. RET/PTC Mutation

Clonal *RET*/PTC rearrangements are the most common genetic alteration found in 20–35% of PTC cases. It is specific for this type of tumor and shows significant geographic variations. *RET*/PTC could also be found in benign thyroid lesions such as thyroid adenomas and Hashimoto thyroiditis. *RET*/PTC rearrangements, especially *RET*/PTC1 and *RET*/PTC3, are more frequently associated with TC following radiation exposure and in children—with or without irradiation history. *RET*/PTC3 correlates with the tendency for aggressive behavior and advanced stage. Positive cases correlate with higher rates of ETE and LNM. The presence of a mutation in the cytological material is strongly predictive for malignancy. The diagnostic and predictive value of *RET*/PTC is controversial. It is still not considered a routine diagnostic tool in cases of indeterminate cytology [44,45].

### 2.6. The Cancer Genome Atlas (TCGA)—DTC Genomic Landscape

The Cancer Genome Atlas data showed that differentiated TC has one of the lowest tumor mutational burdens, and usually harbors only a single “driver” gene alteration. The presence of already known *BRAF^V600E^*, *RAS* mutations and *RET* fusion was described in 59.7%, 13%, and 6.3% of PTC cases, respectively. Research also revealed previously unknown driver genes such as *EIF1AX* (1.5%), *PPM1D* (1.2%), and *CHEK2* (1.2%), thereby decreasing genetically uncharacterized PTC cases from 25% to 3.5%.

Based on gene expression profiles, TCGA introduced a new concept of TC genetic classification. PTC was further divided into two classes that display distinctive differentiation and signaling properties: *BRAF^V600E^*-like (BVL) and *RAS*-like (RL) PTC. 

BVL cancers more likely have either *BRAF^V600E^* mutation or *BRAF*, *RET*, and *NTRK1/3* fusions and show preeminent activation of the mitogen-activated protein kinase (MAPK) signaling pathway. Hence, it was usually associated with the classical subtype of PTC.

RL tumors are predominantly characterized by *H/N/K*-*RAS*, *EIF1AX*, and *BRAF*^K601E^ point mutations and *PPARG* fusions and are activated by both the MAPK and PI3K/AKT signaling pathways [46,47].

In 2016. Yoo et al. suggested a third molecular subtype—Non-BRAF-Non-*RAS* (NBNR), which was mainly associated with follicular-patterned thyroid tumors. NBNR is characterized by alteration in *DICER1, EIF1AX, EZH1, IDH1, SPOP, PAX8-PPARG*, and *THADA* fusion [48].

Each genomic subtype activates different intracellular pathways related to cell proliferation or invasiveness, including the MAPK pathway. The pattern of gradual activation has been noticed: from NBNR to RL to BVL. Only a few NBNR tumors displayed LNM and ETE, unlike the BVL and RL subtypes [49]. 

Awareness and knowledge of DTC genetic features can lead toward more accurate diagnosis and targeted treatments.

## 3. Cell Cycle Control Alteration

Cell cycle proliferation and death must be regulated to maintain tissue homeostasis. The interaction of cell cycle regulators and apoptosis activators define the destiny of the cell. Breakdown of cell proliferation and/or apoptosis control affects the physiology of an organism. Cell cycle dysregulation could potentially lead to different pathological conditions, including cancer development. Understanding the linkage between the cell cycle and apoptosis is crucial for developing new diagnostic methods and therapeutic strategies [50].

### 3.1. Ki-67 

The *Ki-67* antigen, identified by Scholzer and Gerdes in early 1983 [51], is a protein presented during the active phases of the cell cycle. It is absent during the inactive phases. Thus, if present, it is the most valuable marker to evaluate cell proliferative activity. The percentage of tumor cells positive for *Ki-67* nuclear staining is described with the *Ki-67* labeling index (LI). A high *Ki-67* index at the initial diagnosis is usually associated with a less favorable cancer phenotype and a worse prognosis. However, there is lack of standardized cutoff value and the variability of the methodologies. Nonetheless *Ki-67* has been proven to be among most useful prognostic cancer marker in other organ systems. Standardization of terminology reporting thyroid carcinomas regarding *Ki-67* expression risk groups is important to achieve good communication between professionals, and with patients [52,53]. Routine *Ki-67* index determination is not yet a standard practice in thyroid cancer assessment. 

*Ki-67* LI has two potential applications in assessing thyroid lesions: determinations of malignant vs. benign tumors and prediction of progression and outcome [54]. However, in thyroid cancer cases *Ki-67* role is still equivocal. 

The *Ki-67* is a promising marker for differentiation benign tumors from DTC, showing high sensitivity and specificity [55,56,57,58]. If one were to look at FTC or PTC separately, the results are somewhat ambiguous. For example, Aiad et al. suggested that *Ki-67* can differentiate PTC from FTC and even FA, but the *Ki-67* index did not differ if PTC was compared with nodular goiters [57]. Tang et al. found that nodular goiters less intensively expressed *Ki-67* than FA [56]. If comparing PTC variants: aggressive variants were more associated with higher *Ki-67* Li than classic [59]. *Ki-67* expression intensity also tends to be significantly higher in patients with concomitant thyroiditis [56].

Increased *Ki-67* LI is found to be associated with several features recognized as predictors of aggressive potential. One of them is the tumor size [55,56,59,60,61]. Tumors >1 cm express *Ki-67* more intensively than PTMC [56]. Linear correlation is found as well within PTMC subgroups (<5 mm vs. ≥5 mm) [55]. These results are not exactly consistent in all studies [54,62]. *Ki-67* has been additionally associated with depth of thyroid invasion (DTI), i.e., capsular invasion and extrathyroidal extension [49,50,54,56]. In several studies, correlation with metastatic activity was observed, but there was also wide diversity regarding the site of metastases. Zhou et al. reported significant association with the development of central compartment LNM [55], Miyauchi - with lateral compartment LNM, but nothing to do with distant metastases [59]. Matsuse, Ito and Aydogan’s studies resulted in directly opposite findings—no association with the presence of LNM [54,60,62], only with distant spread [54,62].

Increased expression of *Ki-67* was associated with worse disease-free survival (DFS) and cause-specific survival (CSS), suggesting that cell proliferative activity in the primary tumor is a crucial factor representing the risk of persistence/recurrence and mortality [54,56,59,61,62,63].

Preoperative differential diagnosis between benign and malignant tumors, especially FA and FTC, could be challenging and lead to overtreatment or inadequate choice of surgery extent [8]. Classically, *Ki-67* is evaluated in the histological specimens. Ninni Mu et al. study reported that a high *Ki-67* index in FNAB material could be an independent predictive factor for FTC and correlated significantly with the *Ki-67* index of operation tissue specimens. It also allowed predicting extrathyroidal extension and more invasive FTC variants [64].

### 3.2. p27(KIP_1_)

*p27*(KIP_1_) is a factor that restrains cell cycle progression by involving different molecular mechanisms. *p27* functions include coordination of cell proliferation, cell differentiation, and apoptosis [65,66]. In cancerogenesis, *p27* acts as a tumor suppressor. In cancers, synthesis is impaired [67]. Loss of *p27* removes restraints from cyclin-CDKs and promotes tumor progression. *p27* also appears to assume a tumorigenesis stimulating role when localized to the cytoplasm. It promotes cell motility and migration—critical events in cancer dissemination [68,69,70]. *p27* can play both tumor suppressive and stimulating roles, therefore total loss in cancers is rare [71].

Several studies have assessed *p27* expression in thyroid pathologies. It is suggested as a reliable marker to distinguish FA, including atypical types, from FTC. There are also differences in expression between minimally and widely invasive tumors. Loss of *p27* is an early event in follicular carcinogenesis, which starts already in minimally invasive carcinomas [72]. Moreover, significantly decreased *p27* expression (<3.7%) is observed in high-risk FTC patients, compared with an intermediate or low-risk group (according to American Thyroid Association Guidelines [8]). Lower expression is also related to more aggressive phenotype, presence of vascular invasion, and advanced tumor stage. Although low-expressors showed a tendency for worse response to RAI therapy, shorter DFS, and decreased median survival compared with high-expressors (84 vs. 70.3 months), this relation did not reach the point of statistical significance [73].

In PTC, loss of *p27* expression varied from 10.2% to 92.3%. It was noticed significantly more frequently in PTC than in benign lesions. Loss of expression and cytoplasmic localization significantly correlated with aggressive behavior—the presence of LNM and distant metastasis, in some studies also with ETE [74,75,76,77,78,79]. Regarding age, sex, tumor size, growth characteristics of the tumor, and extrathyroidal extension, *p27* showed no significant relation [73].

Frequently cytoplasmic localization of *p27* is noticed in association with a *BRAF^V600E^* positive status [80]. Several studies have identified *p27* downregulation also in non-tumorous thyroid lesions, including Hashimoto thyroiditis and Graves’ disease [75].

### 3.3. Cyclin-D1 

Cyclin-D1 is a nuclear protein that controls cell cycle progression of the G1/S phase. Changes in cyclin expression result in loss of control over normal cell growth and further induces oncogenesis. Overexpression has been noticed in various types of cancers, including TC [79].

Cyclin-D1 is expressed more commonly in malignant than benign thyroid lesions. With some diversities between studies, overexpression was significantly associated with a more aggressive phenotype, including presence of extrathyroidal extension, metastatic disease, risk of recurrence, and lack of response to treatment [81,82,83,84,85,86,87]. Cyclin-D1 with high sensitivity and specificity (85% and 100%) at a certain threshold (46%) of immunolabeled cells could also be used as a cancer predictor in FNAB material. The presence of ETE, intraglandular metastases, and LNM were significant predictors of increased Cyclin-D1 staining [88]. 

### 3.4. Survivin

Survivin is one of the inhibitors of apoptosis protein family members that block cell death through caspase-dependent and independent pathways. Survivin is mainly expressed in cancers. Expression in normal tissue is negligible, therefore it has become a significant diagnostic and prognostic marker, as well as a target for anticancer therapies [89].

Survivin expression in normal thyroid follicles is uneven and mild. It increases along with the dedifferentiation degree [90].

The majority of PTC show moderate to high survivin immunohistochemical staining. Higher expression is observed in tumors with LNM and distant metastases [91,92,93,94], with a tendency for ETE and advanced stage [91,93]. Survivin expression intensity in metastatic tissue is identical to that in initial tumors [95]. Upregulation of expression is favoring tumor progression toward aggressive and poorly differentiated phenotype [96].

### 3.5. P53

The T*P53* gene codes *P53* protein, which stops cell cycle turnover in the G1 phase to repair damaged DNA. The absence of a normal *P53* gene and/or presence of a mutant *P53* gene variant causes alterations in the repair process. In combination with inhibited apoptosis, it results in increased numbers of transformed cells [97]. Tumor suppressor gene *P53* mutation is the most common alteration in malignant cells, represented in approximately half of all human cancers [98].

Several studies have reported the frequent occurrence of *P53* gene mutations in undifferentiated TC. However, the prevalence of alteration in well-differentiated TC, including PTC, has not been clearly established. It ranges from 0% to 25% [97,99]. 

A significantly higher *P53* protein expression level was noticed in malignant lesions than in benign, thereby it could be considered as a diagnostic and prognostic marker for PTC [81,87,98,100]. At the same time, the determination between PTC and FTC was ambiguous [58,100]. The association between *P53* and clinicopathological features of DTC is a matter of controversy. On the one hand, Marcello et al. found a correlation with a patients’ tumor size, multifocality, and metastatic status. On the other, Shin et al. reported no significant correlations [98,100]. *P53* was more frequently associated exclusively with the development of dedifferentiated cancers [101] and *P53*’s role in the prediction of DTC behavior is controversial.

## 4. Estrogen Receptors (ER)

Estrogens are a group of female hormones that play an essential role in reproductive and non-reproductive systems in females and males. In literature, the term estrogen more frequently specifically refers to 17β-estradiol (E_2_) due to its significance [102]. Estrogen functions are mediated mainly through receptors alpha (*ER-α*) and beta (*ER-β*) [103]. Estrogens regulate cell growth, reproduction, development, and differentiation through genomic or non-genomic pathways [102,104]. Abnormalities in E_2_/ER binding, signaling pathways, or imbalance of *ER-α* and *ER-β* contribute to carcinogenesis and cancer progression [103,105,106]. 

The global incidence of TC in women in 2020 was 10.1 per 100,000, i.e., 3-fold higher than it is in men. The high prevalence among women, especially in fertile age and which decreases after menopause, suggests that hormonal and reproductive factors, especially E_2_, can have a significant impact on TC development and progression. [107,108]. The importance of E_2_ in TC occurrence is also indicated by the higher rates of thyroid cancer in women with a history of breast cancer and vice versa [109].

The presence of ER in the thyroid was first reported in 1981 by Molteni et al. Estrogen enhances thyroid cancer cell adhesion, invasion, and migration, due to the downregulation of tumor-suppressive proteins, such as catenins and cadherins. Furthermore, ER acts on genes promoting transcription of angiogenesis factors, such as *VEGF*, and proteolytic enzymes, such as *MMP-2* and *MMP-9* [103,110,111,112]. *ER-α* expression in thyroid cancers is more frequently increased, while *ER-β* is decreased or undetectable [103]. Different ER distribution and expression patterns, as well ERα/ERβ ratio, may have a role in thyroid cancer cell proliferation, development of specific clinicopathological features, and the outcome [105,113,114].

### 4.1. Estrogen Receptor-α (ER-α)

ER could be found in normal and neoplastic thyroid tissue, but the expression patterns are different. More commonly, *ER-α* expression was significantly higher in clinical cancers than in incidental tumors or non-cancerous tissue [115,116,117,118]. On the contrary, Heikkila et al. found no difference between benign and malignant lesions. *ER-α* mainly was negative in both adenomas and carcinomas [119]. Vaiman et al. did not observe immunoreactivity for *ER-α* in thyroid tissue samples at all [120]. It is noticed that incidental cancers are more frequently *ER-α*(-) than *ER-α*(+) [116,117]. No correlation was found regarding age, gender, pre- or post-menopausal status [115,118,121,122]. These data suggest that the intensity of *ER-α* expression may not be the only causative factor for the high incidence of PTC in the female population. Another contradictory opinion was established by Rubio et al., as *ER-α* expression was increased in postmenopausal papillary thyroid cancer cells compared with non-thyroidal controls but was unchanged in premenopausal PTC. At the same time, *ER-β* expression did not change in either group [123].

Regarding tumor clinicopathological features, results vary. In some studies *ER-α* was significantly associated with tumor size [115,121,124]. Different results are presented by Huang et al. -no correlation between tumor size or TNM stage and *ER-α* overexpression was noticed [122]. Magri et al. reported that T1 and T2 tumors with the increased expression of *ER-α* and the loss *ER-β* were associated with a more aggressive tumor phenotype-presence of capsular and vascular invasion. The correlation was lost in T3 and T4 tumors, proposing a minor impact of ER expression on more advanced stages [116]. Rubio et al. suggested that increased *ER-α* expression after menopause may be associated with papillary thyroid cancer aggressiveness [123]. Aggressive behavior and worse overall survival also correlated with increased *ER-α*/*ER-β* in female patients with PTC [118]. 

Considerably different expression patterns of *ER-α* make it difficult to define its true significance in the development and progression of thyroid cancer. However, it is clear that estrogen acting through *ER-α* has an impact on thyroid cell processes. It is proved that ERα positivity was strongly associated with aggressive behavior of DTC. Furthermore, undifferentiated, growth-prone thyroid stem and progenitor cells had 8 to 10 times higher ER expression than differentiated thyrocytes [117,125,126,127].

Sturniolo et al. results, in cont*RAS*t to previous reports, suggest a significant correlation of *ER-α* expression with remission of the disease. *ER-α* positive patients were more commonly disease-free after initial treatment. *ER-α* expression is associated with favorable outcome in PTC patients [124].

### 4.2. Estrogen Receptor-β (ER-β)

*ER-β* is also present in both normal thyroid tissue and cancers. The *ER-β* expression does not depend on E_2_, thereby subsequently also on menopausal status [113,120,123,128]. 

Downregulation of *ER-β* expression, in general, is associated with tumor development and progression [116,117,118]. *ER-β* maintains cell differentiation and the epithelial phenotype. Thereby, decreased or absent expression of *ER-β* may also be a hallmark of TC dedifferentiation [126].

*ER-β*(-) tumors are more commonly presented with vascular invasion than in *ER-β*(+) tumors [117]. Significantly higher expression is observed in tumors without ETE, with co-existing thyroiditis, wild-type BRAF and *RAS* gene mutations, and in tumors with smaller T [118]. Moreover, reduced expression in female patients was associated with increased prevalence of LNM and extra-thyroid spread, especially in combination with *ER-α* negativity. *ER-β* expression was lower in those who required RAI than in the nontreatment group [118,128]. *ER-β* negativity showed tendency for recurrence or persistent disease in young female patients (<45 years old) [129].

In FTC patients, there was no correlation between low *ER-β* expression and gender, age, tumor size, vascular or capsular invasion, metastatic status, site of metastases, proliferative activity, TNM, or stage. Moreover, there was no correlation with tumor differentiation grade, local recurrences, or serum thyroglobulin level before the first RAI therapy episode [119].

Nevertheless, it is clear that estrogen has an important role in thyroid cancer development, especially in females. Results frequently show a large diversity of ER expression patterns [119]. Thereby, over- or under-expression of ER cannot be used as the only predictive marker to determine cancer diagnosis, prognosis, or outcome. ER should be combined with other markers [119,130]. 

## 5. Epithelial-Mesenchymal Transition (EMT)

An epithelial-mesenchymal transition is a complex developmental process that allows polarized epithelial cells to suppress their epithelial features and change to mesenchymal, assuming the qualities of mesenchymal cells: enhanced migratory capacity, invasiveness, and higher resistance against apoptosis [131]. EMT was first discovered by Elizabeth Hay in 1960 [132]. It is a physiological process necessary for normal embryologic development and wound healing. In the case of pathology, signaling pathways are different from those in normal tissue, causing abnormal transition [133]. Pathological EMT is provided by a complex and coordinated set of molecular changes, leading to a favorable microenvironment for cancer progression and metastases formation [131,134,135]. These qualitative and quantitative changes can be used for the evaluation of cancer development and progression [136]. 

The current understanding of the role of the epithelial-mesenchymal transition in TC is that the EMT is responsible for the progression from DTC to poorly differentiated thyroid carcinoma and anaplastic thyroid cancer [137].

### 5.1. E-Cadherin

*E-Cadherin* is predominantly detected in epithelial cells. Decreased *E-Cadherin* expression reduces cell-cell connections and induces EMT processes, resulting in enhanced tumor invasiveness [137,138,139].

It is expressed in normal thyroid tissue and benign lesions, while its expression reduces with the progression of differentiated TC towards undifferentiated ones. Loss of *E-Cadherin* expression is noticed in the majority of anaplastic thyroid cancers [5,140,141,142,143,144,145]. 

There are inconclusive reports regarding *E-Cadherin* expression in PTC compared with FTC, but a significant correlation was noticed regarding PTC subtypes [142,143,144,146]. No significant differences were found in relation to age, gender, or ETE [142,146]. Contradictory results were reported regarding the metastatic status and tumor stage. Several studies found no association, other that *E-Cadherin* expression was significantly downregulated in cancers with aggressive phenotype [138,142,143,147]. In both groups no correlation with the presence of distant thyroid cancer metastasis was found. Regarding prognosis, *E-Cadherin* seems to be an unreliable predictor of DFS [148].

### 5.2. Vimentin

Vimentin is a significant component of the intermediate filament family of proteins, known to coordinate interactions between intermediate filaments and other cytoskeletal elements and organelles. The increased expression has been reported in various epithelial cancers, including gastrointestinal tumors, prostate cancer, breast cancer, CNS tumors, TC, etc. The exact role of vimentin in cancer progression remains unclear, but increased expression could be associated with increased tumor growth, invasion, and poor prognosis [5,137]. In adults, vimentin expression is limited to connective tissue mesenchymal cells, muscle cells, and CNS [149].

Different immunostaining patterns have been identified among thyroid cancer—basal subnuclear in most papillary carcinomas and diffuse cytoplasmic in follicular carcinomas—but differences are insignificant. Vimentin expression levels in FTC and PTC, including PTC subtypes, did not show relevant differences [142]. However, vimentin was more frequently overexpressed in anaplastic cancer than in PTC [150]. Calangiu et al. found no significant association of vimentin immunostaining and gender, age, presence of LNM, or tumor stage [142]; Park et al. reported that expression was increased in PTC with LNM [150]. Yamamoto et al. found the opposite tendency—weaker and more focal vimentin expression in PTC with distant metastases than those without metastases or even occult cancers [151]. Vimentin positivity was significantly associated with a reduced DFS [148]. Although vimentin’s role in thyroid cancer development and progression is not widely described and frequently results are contradictory, it is associated with poor prognosis in other tumors.

### 5.3. N-Cadherin

*N-Cadherin* is a cell adhesion molecule playing an important role in EMT. A common feature of EMT is the so-called “cadherin switch”, the loss of *E-Cadherin* expression and the concomitant upregulation or de novo expression of *N-Cadherin*. This process is associated with increased migratory and invasive ability and a worse patient prognosis [152,153].

*N-Cadherin* expression level was significantly higher in PTC tissue than in normal para-carcinoma and normal non-cancerous thyroid tissue. [154,155]. *N-Cadherin* stimulates thyroid tumorigenesis through enhancing cell proliferation, colony growth and inhibiting cell apoptosis. Additionally, *N-Cadherin* promotes the metastatic potential of the tumor through extracellular matrix degradation processes, which increased *MMP-2* and 14 expression, and the transcription of genes associated with induction of metastatic activity [155]. Significant differences were found between patients with and without LNM. No correlation was found between expression and age, or gender of PTC patients, as well tumor size [154].

*N-Cadherin* possesses the ability to affect the cytoskeleton, cross-talk with other membrane receptors, and mediate cell adhesion, suggesting that the increased activity of *N-Cadherin* may facilitate the development of cancer and gain more aggressive features. *N-Cadherin* has an important role in the pathogenesis and progression of multiple cancer types, but its exact role in thyroid cancer and whether it can be used as a biomarker is still undetermined and requires more studies [152,153].

### 5.4. CD44

*CD44* is a universal surface molecule that exists as several isoforms and participates in multiple physiological processes. Abnormal expression and dysregulation of *CD44* contribute to tumorigenesis initiation and progression. *CD44* represents a common biomarker of cancer stem cells and promotes EMT. It is involved in cancer proliferation, invasion, metastatic activity, and resistance against therapy. In addition, *CD44* can serve as an adverse prognostic marker among the cancer population and target for therapeutic intervention [156,157,158]. Aberrant expression and/or dysregulation of *CD44* facilitate tumor formation in multiple organs, including lungs, ovaries, liver, glioma, and thyroid gland [156].

*CD44* variant 6 (*CD44*v6) is expressed only in proliferating thyroid cells, and it is upregulated in carcinomas. In addition, upregulation is linked to aggressive behavior and the occurrence of metastasis. 

The immunohistochemical score of *CD44*v6 protein staining is significantly higher in PTC specimens than in benign thyroid nodules or adjacent normal thyroid tissue [159,160]. PTC with lateral LNM showed more intensive *CD44*v6 staining than without metastases. High levels of *CD44*v6 protein in PTC specimens may aid in the prediction of disease progression in early-stage patients. It has the potential to be a predictive marker for LNM. If present, individualized treatment can be initiated as early as possible [157]. In another study, *CD44* (not specifying isoform) alone did not show a statistically significant association with age, gender, tumor size, LNM status, multifocality, extrathyroidal extension, or recurrence. However, there was a correlation with the combination of *CD44*(+) and *CD24*(-). Particularly, the combination of *CD44*+/*CD24*- demonstrated a significant association with age and gross ETE. *CD44*(+)/*CD24*(-) was also identified as an independent prognostic factor for PTC. Combinations exhibited a statistically significant negative association with DFS [161].

## 6. Extracellular Matrix Degradation

Tumor invasion is a dynamic, complex, multi-step process, which involves the detachment of malignant cells from their point of origin, migration through extracellular matrix and basement membranes, and invasion into lymphovascular channels. Breakdown or proteolytic degradation of the basement membrane components and extracellular matrix are essential steps for invasion. It is provided by specific proteases [162]. Matrix metalloproteinases (MMPs), a family of zinc-dependent endopeptidases, were first described by Gross and Lapiere in 1962 [163]. Particularly *MMP-2* and *MMP-9* are important factors of basal membrane degradation and the matter of interest in cancer research [164,165]. MMPs influence diverse physiologic and pathologic processes [166]. The function depends on the local balance between MMPs and their physiological inhibitors. MMP expression is upregulated in almost every type of human cancer, and their expression is often associated with poor survival. Whereas some of the MMPs (e.g., MMP-7) are expressed by the cancer cells, other MMPs (*MMP-2* and *MMP-9*) are synthesized by the tumor stromal cells, including myofibroblasts, fibroblasts, endothelial cells, and inflammatory cells [163].

### 6.1. MMP-9

*MMP-9* is a member of the gelatinase group, also known as gelatinase/type IV collagenase. It digests denatured collagens and gelatins and also plays a particular role in angiogenesis by increasing the activity of proangiogenic factors [165]. Upregulation of *MMP-9* has been shown to facilitate cancer progression, invasion, and development of metastases in various types of human malignancies [166].

*MMP-9* expression is found to be greatly upregulated in thyroid carcinomas. Several studies reported *MMP-9* overexpression in samples obtained from patients with PTC, compared with benign thyroid nodules or normal thyroid tissue [167,168,169,170,171]. MMPs are involved in the stimulation of angiogenesis. It is essential for tumor growth and progression [172,173], therefore, *MMP-9* expression is significantly higher among PTC patients with larger tumor size [168,174], especially in tumors >2 cm. It is considered a dependent predictor for disease status and DFS, indicating that *MMP-9* correlates with the PTC prognosis [167,168,169]. Higher intensity of *MMP-9* expression is observed in patients with LNM (central and/or lateral) [167,168,172,174,175,176,177]. Different results were documented by Meng et al., who found no significant correlation between *MMP-9* and lymph node metastasis [169].

Šelemetjev et al. found a significant correlation with the degree of tumor infiltration [172]. On the contrary, Liu et al. and Zarkesh et al. groups did not find any correlation with the presence or absence of capsule invasion, multifocality, extrathyroidal invasion, and distant metastases [167,171]. Despite similarities with the study results by Liu et al., Zarkesh et al. also reported *MMP-9* association with vascular invasion, as well as the absence of correlation with *BRAF^V600E^* mutation [171]. The majority of studies revealed a positive correlation between *MMP-9* expression and advanced tumor stage [164,167,172]. In cont*RAS*t, Wang et al. found no significant correlation with the advanced PTC stage [168].

When comparing expression patterns of both active and total *MMP-9*, active *MMP-9* was presented almost only in cancer tissue and correlated with age, LNM, and ETE. On the other hand, total *MMP-9* expression had no significant correlation with any clinicopathological feature, suggesting that *MMP-9* is activated in tumor cells. Therefore, it is associated with aggressive tumor behavior in PTC [174]. During the follow-up, patients with a persistent disease status had a significantly higher *MMP-9* score (multiplication of intensity and percentage of positively stained cells) than with a disease-free status. DFS time was shorter in PTC patients with higher *MMP-9* scores than with lower *MMP-9* scores [167].

*MMP-9* could be detected not only in thyroid tissue but also in peripheral blood in patients with normal thyroid, benign pathologies, and thyroid cancer. Data from peripheral blood studies correspond to thyroid tissue study data. The levels of *MMP-9* in the peripheral blood of patients with DTC are significantly higher than in those with benign thyroid disease [178,179,180]. No difference is found between patients with benign thyroid disease and healthy volunteers [179,180]. It was observed that *MMP-9* levels in the peripheral blood pre- and post-operatively did not significantly change in benign lesions, whereas in patients with DTC, post-operative *MMP-9* levels were significantly lower than pre-operative. In DTC with a higher TNM stage, tumor diameter ≥l cm, extrathyroidal extension, or existing lymph metastasis and distant metastases, serum *MMP-9* levels were significantly higher than those reported in tumors with early TNM stage and smaller diameter [178,180].

### 6.2. MMP-2

*MMP-2*, also known as Gelatinase A, can degrade type IV collagen. Upregulation of *MMP-2* is suggested to be associated with the malignant behavior of tumor cells, including in PTC.

The immunohistochemical score for *MMP-2* protein staining was significantly higher in PTC specimens than in benign thyroid nodules and adjacent normal thyroid tissue. The results show that the increased intensity of *MMP-2* staining in PTC specimens significantly correlated with ETE and LNM and distant metastases [159,181]. The study results by Saffar et al. suggest no relation with capsular invasion and only borderline correlation with vascular invasion [182].

The study results of *MMP-2* expression in thyroid tissue frequently correlated with peripheral blood level evaluation. *MMP-2* concentration was higher in the PTC group than in the benign thyroid nodule group or the healthy control group. *MMP-2* expression above a certain cutoff value could help to differentiate between PTC and benign thyroid tissue. Preoperative serum *MMP-2* has a moderate predictive capacity for LNM. High *MMP-2* levels in the blood (>86.30 ng/mL) were associated with more significant risks for the development of larger tumors (>1 cm), CLNM, LLNM, extrathyroidal invasion, and advanced TNM stage. *MMP-2* levels negatively correlated with time till recurrence. High serum *MMP-2* level was associated with a worse clinical outcome [183].

The serum concentrations of *MMP-2* and *MMP-9* differ significantly regarding age, presence of microcalcification, irregular shape, diameter or number of cancer foci, and other clinicopathological features mentioned earlier [184].

Several studies suggest that levels of *MMP-2* and *MMP-9* in the peripheral blood could be used not only as predictive markers of DTC [178,179,184] but also as the efficacy criterion for different treatment methods, e.g., surgery [168] or minimally invasive procedures, such as Ult*RAS*ound-guided Radiofrequency Ablation (RFA) [184]. Serum levels of *MMP-2* and *MMP-9* could provide valuable references for diagnosing PTC before RFA and information about procedures efficacy. In cases of DTC, *MMP-2* and 9 levels were significantly lower after RAF than before the procedure. No changes were noticed in the benign nodule group. In combination with relevant risk factors, these serological indexes may help to predict the prognosis of PTC before and after ablation or surgery and could have significant implications for the PTC management planning [178,184].

## 7. Hypoxia End Neovascularization

Hypoxia is one of the most characteristic processes of cancer development and progression. Intensive proliferation and expansion of the tumor increase oxygen demand, which is not always equal to supply [185], leading to self-sustaining processes such as ischemia, neovascularization, growth, and cancer progression.

### 7.1. Hypoxia-Induced Transcription Factor

Hypoxia is a prominent micro-environmental component in many types of solid tumors due to inadequate vascularization [186]. The cellular response to hypoxia is mediated by the hypoxia-inducible transcription factors *HIF-1* and HIF-2.

Activated Hypoxia-inducible factor 1 (*HIF-1*) regulates the responses of the tumor cells to changes in oxygen concentration through transcriptional activation of genes [187,188,189]. *HIF-1* transcriptional induction of proangiogenic factors, e.g., the vascular endothelial growth factor (*VEGF*), stimulates new blood vessels development, thereby facilitating tumor cells supply with oxygen. In addition, *HIF-1* stimulates tumor metastatic activity—the migration of cancer cells into distant and more oxygenated tissues through the transcriptional activation of oncogenic factors [190,191]. *HIF-1* complex, consisting of two submits, the oxygen-sensitive *HIF-1*α and the constitutively expressed *HIF-1*β, participates in the cancer biology of numerous endocrine tumors [192]. *HIF-1*α and HIF-2α have unique tissue distributions and play critical but non-overlapping roles in tumor progression [187,193].

Cancer development and progression depend mainly on three factors—the presence of hypoxia (*HIF-1*a activation), inflammation (NF-kB activation), and *ER-α* [103].

Several studies reported higher expression of *HIF-1*α and HIF-2α in TC than in normal thyroid or benign lesions [194]. The expression did not show any correlation with the patient’s age, gender or the presence of node calcification. However, there was a strong correlation between HIF-2α and tumor size; it did not reach the same significance in association with *HIF-1*α. *HIF-1*α and 2α overexpression was associated with capsular invasion and the presence of LNM. Tumors with high *HIF-1*α and 2α staining had a higher TNM stage [194]. Data suggest that *HIF-1* may promote migration and aggressiveness of PTC, FTC, and ATC [195]. *HIF-1* knockdown represses cell invasion and induces apoptosis by downregulating the expression levels of WWP9, WWP2, *VEGF*, and *VEGF*R2. Therefore, *HIF-1*α could serve as a potential therapeutic target for the treatment of TC [196].

### 7.2. Vascular Endothelial Growth Factor (VEGF)

The vascular endothelial growth factor is the key mediator of cancer angiogenesis. It is upregulated by oncogene expression, various growth factors, and also hypoxia [197]. Vascular endothelial growth factors are members of the platelet-derived growth factor family of structurally related mitogens. This group of markers includes *VEGF*-A, placental growth factor (PlGF), *VEGF*-B, *VEGF*-C, and *VEGF*-D [198].

*VEGF* has been identified as a significant mediator of angiogenesis in the thyroid gland. Angiogenesis is known to play an important role in the development, growth, and metastasizing of cancers. 

*VEGF*-C is the most potent factor, which guides the growth of tumor-associated lymphatic vessels, thereby further promoting cancer spread through the lymphatic system [199]. Ceric et al. found no difference between papillary and follicular cancer [200]. However, in another study, *VEGF*-C differed significantly between PTC histotypes (classic vs. follicular variant). *VEGF*-C did not differ between age groups and sex [95]. *VEGF*-C positively correlated with tumors T status—the higher the T stage, the more intensive of an expression of *VEGF*-C was observed. *VEGF*-C also correlated with gross ETE, and positive resection margins [190]. A high expression was associated with the presence of lymph node metastases and advanced TNM stage [95,172,200,201]. Another study compared *VEGF*-A and *VEGF*-C, and their co-regulatory potential in different types of thyroid carcinomas. *VEGF*-A overexpression was observed in undifferentiated cancers, suggesting its important role in developing aggressive patterns. Diversities related to association between both *VEGF* types and development of LNM were found. *VEGF*-A was highly expressed in classical variant of PTC with LNM, but *VEGF*-C was significantly under-expressed. *VEGF*-C expression was higher in follicular variant of PTC. Results show that *VEGF*-A and *VEGF*-C mRNA and protein productions in thyroid malignancies have a strong association and may cross-regulate each other. It could be related to their shared receptor (*VEGF*R2) and resultant feedback systems. It has been noticed that *VEGF*-C can stimulate angiogenesis in addition to lymphangiogenesis [202]. The upregulation of *VEGF*s in human TC correlates with cancer development, progression, and poor prognosis [187,203].

### 7.3. Antiangiogenic Therapy—Tyrosine Kinase Inhibitors

DTCs are highly vascular tumors, meaning that they are dependent on angiogenesis. Angiogenesis and cancer cell growth are influenced by aberrations in cancer cell signaling. Most of these aberrations occur as a result of mutations in tyrosine kinase and other molecule-affected pathways (RAF/MEK/ERK and AKT/mTOR).

In recent years, angiogenesis has been intensively studied regarding the identification of mediators that may be used as targets for new anticancer treatments.

Multiple Tyrosine kinase inhibitors (TKI) targeting RAF, MEK, and the *VEGF* receptors have been identified. TKIs limit tumor growth by decreasing their blood supply. In addition, TKI have direct anticancer activity on the tumor cells by targeting signaling molecules within or upstream of the RAF/MEK/ERK pathway. Their efficacy on different molecular pathways in case of recurrent/metastatic RAI refractory TC has been researched [204,205]. 

Several TKIs have been approved for the treatment of advanced DTC. Sorafenib is approved for the treatment of progressive metastatic RAI refractory DTC. It inhibits *VEGF*1–3, platelet-derived growth factor (PDGF), fibroblast growth factor (FGF), KIT, and *RET* receptors and weakly RAF kinase. The response to treatment was observed in 12.2% of patients. 

Another TKI, Lenvatinib, was approved in 2015 for treatment of advanced RAI refractory DTC. Lenvatinib targets *VEGF*R 1, 2, and 3, fibroblast growth factor receptors (FGFR) 1 through 4, PDGFR α, *RET*, and KIT signaling networks. It was associated with significant improvements in progression-free survival and the response rate among patients with RAI refractory TC. The response to Lenvatinib therapy was observed in 64.8% of cases [204,205,206]. 

Nevertheless, TKI therapy is associated with significant improvements in progression-free survival and the response rate among patients with treatment refractory DTC, patients can experience adverse effects and impaired quality of life, as well substantial financial burden [206]. 

## 8. Aggressive Variants of PTC

In addition to knowledge of the genetic and molecular impact on cancer aggressive behavior, the role of specific morphological entities of PTC should also be recognized.

PTC usually is associated with a good prognosis and overall survival. Aggressive behavior could be associated with the presence of rare but more aggressive PTC subtypes such as diffuse sclerotizing variant, hobnail variant, tall cell variant, columnar variant, or solid/trabecular variant of PTC.

Tumors harboring aggressive variants are associated with advanced stage, extrathyroidal extension, higher recurrence rate and decreased disease-specific survival compared to the classical or follicular variant of PTC. The development of these tumors showed a high correlation with gene and molecular mutations such as *BRAF^V600E^*, *RAS*, *TERT*, *RET*/PTC3, and T*P53*. For a comprehensive assessment of DTC to predict prognosis and formulate a more complete treatment plan, these PTC variants should be considered and recognized [207,208].

## 9. Conclusions

The incidence of thyroid cancer (TC), in particular differentiated thyroid cancer (DTC), has been increasing over the last few decades. Regarding behavior, TC can be stratified as low, intermediate, and high risk, depending on a variety of morphological, genetic and molecular features. The traditional “one-size-fits-all” approach to DTC should be revised and transitioned to a more individualized and risk-adapted approach for the selection of more appropriate initial therapy, adjuvant therapy, and follow-up of patients with DTC. Therefore, early recognition of potentially worse cancer patterns can lead to more timely risk assessment and prediction of adverse events, which can potentially impact DTC patients’ destiny. 

A large amount of studies has been conducted for the last few decades regarding TC risk assessment. DTC behavior, recurrence, and survival could be predicted by analyzing genetic and molecular markers. Nonetheless, so far, no consensus concerning the clinical prognostic value of studied markers have been achieved.

After summarizing data from analyzed studies, we conclude that there is a large amount of different genetic and molecular markers for the detection of the possible impact on tumorigenesis, aggressive phenotype, recurrence, and overall survival. Therefore, we cannot highlight one single marker which could accurately predict the possible behavior of DTC. Frequently diverse and even contradictory results are noticed among studies. If speculating on possible reasons, the heterogeneity of cohorts (size and characteristics) and large diversity of methodologies should be mentioned. One more reason could be the unequal numbers of research dedicated for the evaluation of different markers. To determine the true significance of each marker and their role in behavior of DTC, implication of more and comparable equally designed studies could be suggested.

There are inconsistencies due to the high diversity between study results, but some trends could still be observed. (See in Table 1).

Most likely this field of research will be topical in future, due to the increased demand for more accurate and individualized risk assessment and treatment.

## Figures and Tables

**Table 1 medicina-57-01131-t001:** Association of genetic and molecular markers with clinicopathological Differentiated thyroid cancer (DTC) features and outcome.

	Age	Gender	Tumor Size	Extrathyroidal Extension (ETE)	Multifocal Tumors	Lymph Node Metastases (LNM)	Distant Metastases	Stage	Recurrence	Worse Survival
*BRAF^V600E^*	+/-	+/-	+/-	+/-	+/-	+/-	+/-	+/-	+/-	+
*TERT*	+	+	+	+/-	-	+/-	+	+	+	+
*BRAF^V600E^* + *TERT*	+	-	n/a	+	+	+	+	+	+	+
*RAS*	n/a	n/a	n/a	n/a	+	n/a	+	n/a	+/-	+/-
*RET*/PTC	n/a	n/a	n/a	+	n/a	+	n/a	n/a	n/a	n/a
*Ki-67*	+/-	-	+/-	+	-	+/-	+/-	+	+/-	+
*p27*	-	-	+/-	+/-	-	+/-	+	+	n/a	n/a
Cyclin-D1	-	-	+/-	+/-	-	+/-	n/a	+/-	+/-	+
Survivin	-	-	-	+	n/a	+	+	+	n/a	n/a
*p53*	-	+/-	+/-	+/-	+/-	+/-	+	-	-	n/a
*ER-α*	-	-	+/-	-	-	-	-	-	-	+
*ER-β*	-	-	-	+ *	-	+ */-	-	-	-	+
*ER-α*/*ER-β*	+	-	-	-	-	-	-	-	n/a	+
*E-Cadherin*	-	-	+	+/-	n/a	+/-	-	+/-	-	n/a
Vimentin	-	-	n/a	n/a	n/a	+/-	-	-	+	n/a
*N-Cadherin*	+	-	+	n/a	n/a	+	n/a	n/a	n/a	n/a
*CD44*	+ **/-	-	-	+ **/-	-	+ **/-	n/a	n/a	+ **/-	n/a
*MMP-9*	+/-	-	+/-	+/-	-	+/-	-	+/-	+	+
*MMP-2*	-	-	+/-	+	-	+	+/-	+	+	n/a
*HIF-1*	-	-	-	n/a	n/a	+	n/a	+	n/a	n/a
*VEGF*	-	-	+	+	-	+/-	n/a	+	+	n/a

“+/-”—contradictive data. “n/a”—data not available. “+”—correlation found. “-”—no correlation found. *—only in papillary cancer (PTC). ** +*CD24*(-).

## Data Availability

Not applicable.

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
