# Peer review of "Can We Predict Differentiated Thyroid Cancer Behavior? Role of Genetic and Molecular Markers"

_medicina, 2021, doi:10.3390/medicina57101131_

Round 1
Reviewer 1 Report
In the manuscript, the authors describe genetic and molecular markers that may predict well differentiated thyroid cancer prognosis. The information will attract attention from breast cancer clinicians and researchers. There are, however, some concerns as follows.
- Morphological or histopathological aspects cannot be ignored. Extrathyroidal extension (ETE) and vascular invasion are histopathologically determined. Certain histological variants, such as papillary carcinoma tall cell variant and hobnail variant, are reported to show rather worse prognosis (Nath and Erickson, Adv Anat Pathol 25:172-179, 2018).
In addition, some of the molecular markers, such as cyclin-D1 and P53, seem to be closely related to histological diagnosis of the poorly differentiated carcinoma or the anaplastic carcinoma. These markers are not appropriate for the predictive markers of well differentiated papillary/follicular carcinoma.
- As to the genetic markers, the authors should describe the genomic landscape of papillary carcinoma defined by the TCGA (Cell 159: 676-690, 2014).
- As to the Estrogen receptors (ERs), the authors should mention the importance of E2 levels of the patients. In consistence, premenopausal female patients (<45 years old) have better prognosis. They might discuss the effect of ERs in pre and postmenopausal patients, separately.
- As to the hypoxia and neovascularization, the authors should discuss the effects of the kinase inhibitors, such as lenvatinib and sorafenib. They are reported to inhibit neovascularization by inhibition of VEGF etc.
- p.7, line313. “negatively positively” should be “negatively”.
Author Response
Dear Reviewer,
answers to Your comments are in added document.
Kind regards,
Rita Niciporuka

Reviewer 2 Report
The authors described extensively several genetic and molecular markers for differentiated thyroid cancer. Main conclusion is that they can contribute to prediction of disease development, progression, and prognosis.
After reading the manuscript some important issues remain:
- Abstract
See my mark in the conclusion section.
- Introduction
Going from the Introduction towards Section 2 (‘Genetic aberrations markers’), I miss a section describing the structure of the paper. It would be easier for the reader if you give some introduction on what will be discussed and in which sequence.
- Conclusion
- In previous sections different markers were extensively discussed. Therefore, it would be helpful to have a combination of a summary and a conclusion for the most important markers. Currently, the conclusion section is more like a general conclusion which also could have been drawn without the previous sections. For example, Line 630-632 (‘The vast majority of studies acknowledge the role of genetic alterations in predicting aggressive tumor behavior and prognosis’); so which markers and is it regarding survival, recurrence, or another outcome. So please change to conclusion section in a way that it is more tailored to your research question about can genetic and molecular markers predict DTC behavior, i.e. which ones and in which way.
- In the Abstract (Line 27/28) it is stated that large randomized studies are required to evaluate the true significance of predictive markers. Besides that I wonder how this would be ever possible (regarding study duration and sample size), as this is apparently an important conclusion of the manuscript, it would be helpful that your view on this is also discussed in the conclusion section.
- General comment
It would be very helpful to have a table with the major learning points/conclusions from the current review
Author Response
Dear Reviewer,
Answers to Your comments are in added document.
Kind regards,
Rita Ničiporuka

Round 2
Reviewer 1 Report
My concerns are appropriately addressed.
The manuscript will attract attention from thyroid cancer researchers.
Reviewer 2 Report
Thank you for revising the manuscript and the answering the raised questions. I’m satisfied with the answers and especially like the added table.
This manuscript is a resubmission of an earlier submission. The following is a list of the peer review reports and author responses from that submission.